# Optimized Visual Internet of Things for Video Streaming Enhancement in 5G Sensor Network Devices

**DOI:** 10.3390/s23115072

**Published:** 2023-05-25

**Authors:** Anil Kumar Budati, Shayla Islam, Mohammad Kamrul Hasan, Nurhizam Safie, Nurhidayah Bahar, Taher M. Ghazal

**Affiliations:** 1Institute of Computer Science & Digital Innovations, UCSI University, Kuala Lumpur 56000, Malaysia; 2Center for Cyber Security, Faculty of Information Science and Technology, Universiti Kebangsaan Malaysia, Bangi 43600, Malaysia; nurhizam@ukm.edu.my (N.S.);; 3Applied Science Research Center, Applied Science Private University, Amman 11937, Jordan; 4School of Information Technology, Skyline University, Sharjah P.O. Box 1797, United Arab Emirates

**Keywords:** Visual Internet of Things, visual sensor, video streaming, video compression, 5G networks

## Abstract

The global expansion of the Visual Internet of Things (VIoT)’s deployment with multiple devices and sensor interconnections has been widespread. Frame collusion and buffering delays are the primary artifacts in the broad area of VIoT networking applications due to significant packet loss and network congestion. Numerous studies have been carried out on the impact of packet loss on Quality of Experience (QoE) for a wide range of applications. In this paper, a lossy video transmission framework for the VIoT considering the KNN classifier merged with the H.265 protocols. The performance of the proposed framework was assessed while considering the congestion of encrypted static images transmitted to the wireless sensor networks. The performance analysis of the proposed KNN-H.265 protocol is compared with the existing traditional H.265 and H.264 protocols. The analysis suggests that the traditional H.264 and H.265 protocols cause video conversation packet drops. The performance of the proposed protocol is estimated with the parameters of frame number, delay, throughput, packet loss ratio, and Peak Signal to Noise Ratio (PSNR) on MATLAB 2018a simulation software. The proposed model gives 4% and 6% better PSNR values than the existing two methods and better throughput.

## 1. Introduction

Over the last two decades, the development of the Internet of Things (IoT) has demonstrated its excellent conditions, and it has the potential to be outfitted with “smart” things enriched with modern digitalization, allowing for flawless internet collaboration. This Internet of Things concept has progressed from modest and personal implementations to scale-integrated operations worldwide. In the late 1990s, Radio Frequency Identification (RFID) devices were popular for supporting various Internet-oriented applications. The worldwide flourishing of the Internet of Things (IoT) in the past decade has enabled numerous new applications through the internetworking of various devices and sensors. More recently, visual sensors have been deployed widely for various IoT applications because these sensors can provide more affluent and versatile information. The Internet of Video Things (IoVT) has been considered a large-scale visual sensor. Moreover, the IoVT has unique sensing, transmission, storage, and analysis characteristics that fundamentally differ from the conventional IoT. These new characteristics of IoVT are expected to impose significant challenges to existing technical infrastructures. This article investigates the depth of understanding of recent advances in the various fronts of IoVT and addresses a broad range of challenges [1].

Presently, internet traffic has increased due to multimedia streaming at home or in the workplace after the global pandemic threat. Due to this traffic, there is tremendous pressure on the network infrastructure. Massive bandwidth is required to manage this internet traffic, which is also needed to support the multicasting technique with the support of IoT. The authors proposed software-defined NG-EPON for adaptive video data streaming using 5G technology. This video-streaming quality of service can be achieved at a high level by the incorporation of a Dynamic Bandwidth and Wavelength Allocation (DWBA) scheme [2].

Visual sensor networks (VSNs) are widely used in smart homes, object recognition, and wildlife surveillance. Scalar sensors produce much less data than visual sensors do. These data are difficult to store and send. One popular video compression standard is high-efficiency video coding (HEVC/H.265). HEVC, which can compress the visual data with a high compression ratio but leads to high computational complexity, reduces the bit rate by about 50% compared to H.264/AVC while maintaining the same level of video quality. This paper suggests an H.265/HEVC accelerating method with excellent efficiency and hardware friendliness to overcome this complexity for visual sensor networks. The recommended method uses texture direction and complexity to accelerate intra-prediction for intra-frame encoding and skip redundant processing in the CU partition. Experimental findings showed that the suggested approach could decrease [3].

The effective transmission of video streams is essential because video has overtaken other types of traffic as a significant kind of traffic through wired and wireless networks. Therefore, maximizing available bandwidth for wireless networks while maintaining user requirements for Quality of Service and Quality of Experience is vital. Accurately predicting future video frame sizes can be very helpful in achieving this goal. This research aims to produce such a precise prediction for videos encoded with H.264 and H.265, the two leading modern standards according to market share. Unlike other studies, we model the changing bit rate video traces using neural networks and employ single-step and multi-step methods to capture the long-range and short-range dependent features. We assess the convolutional long short-term memory’s accuracy [4].

Real-time multimedia applications today demand good video quality at acceptable bitrates. Compared to other coding schemes, the H.264 coding provided low bitrate costs while maintaining excellent video quality, but it was limited to higher rates. Later, by offering higher video quality at an effective bitrate, High-Efficiency Video Coding (HEVC) improved on H.264. This enhancement requires increased processing costs due to improved methods such as quad-tree for CTU partitioning. The framework for CTU partitioning proposed in this paper, CtuNet, is approximated using deep learning methods. To forecast the CTU partition of the HEVC standard, a ResNet18-CNN model is used. We used cutting-edge techniques to establish a baseline for our recommendation. The outcomes show how superior the suggested CtuNet is to the competing strategies [5].

The video has become crucial for monitoring, identification, and knowledge sharing. Videos encoded using particular protocols are transferred to intelligent gateways for industrial applications, namely the Industrial Internet of Things (IIoT). In a typical IIoT scenario, the video protocol is initially identified to set up future video tasks. It is difficult for IIoT because of the limited resources in such systems; the video quality can deteriorate during encoding and compression procedures. Deep neural networks (DNNs) have recently been the focus of substantial studies concentrating on protocol identification (PI) and video quality enhancement (VQE) activities on IIoT edge devices. DNNs frequently need a lot of computing power; sophisticated networks are hardly ever deployable on edge devices [6].

To meet the demand for multimedia services, video transmission in the IoT system must ensure video quality while lowering the packet loss rate and latency with limited resources. This research suggests an energy-efficient IoT video transmission technique that guards against interference based on reinforcement learning. The encoding rate, modulation and coding system, and transmit power of the IoT device are all under the control of the base station. Without knowledge of the transmission channel model at the transmitter and receiver, the state-action-reward-state-action reinforcement learning algorithm selects the transmission action based on the observed state. The authors suggested an energy-efficient IoT video transmission technique based on deep reinforcement learning that uses a deep neural network to approximate the Q value to speed up the learning process for selecting the best transmission action and to enhance video transmission performance. A summary of the literature is shown in Table 1 [7].

A 5G network can offer high throughput with low latency by connecting many devices using a IoT-based network infrastructure. Nowadays, the 5G-enabled IoT gives service to smart cities, healthcare, defense, education, etc. One of the biggest tasks of the 5G-enabled IoT is to mitigate the service response time with computational techniques. The authors proposed the edge computing technique with the IoT for smart city users [8].

Compared to 5G, 6G networking is considered to show much better performance with respect to reliability parameters, low delay and high bandwidth. Furthermore, 6G technology is developed by migrating space, cellular, and underwater networks [9].

**Table 1 sensors-23-05072-t001:** Summary of literature survey.

Author’s Names	Methods Used	Limitations
Chen C.W. et al. [1]	Large-scale visual sensor-based Internet of Video Things, Dynamic Bandwidth and Wavelength Allocation (DWBA) scheme	Due to Bandwidth constraints, the packet loss is greater, and the transmission delay is greater
Ganesan, E. et al. [2]	5G-enabled adaptive, scalable video streaming multicast in a software-defined NG-EPON network	Quality of service parameters performance is poor
Ni, CT et al. [3]	HEVC/H.265, H.264/AVC are used for video compression	50% of video quality is lost because of H.264, and 45.33% of Video Quality is lost because of H.265
Om, K. et al. [4]	Accuracy is estimated by applying long short-term Memory, Convolutional Neural Networks and Sequence-to-Sequence models to H.264 and H.265	Poor accuracy is obtained because of the conventional neural networks
Zaki, F et al. [5]	CtuNet framework-based H.264 and H.265 protocols are used	Because of the low data rate, the transmission delay is increased to a greater extent
Chen, L et al. [6]	Protocol identification (PI) and video quality enhancement (VQE) tasks on IIoT edge devices using deep neural networks (DNNs)	97% accuracy is achieved, but latency is increased
Xiao, Y et al. [10]	The reinforcement learning algorithm is used for video streaming in IoT	Increases the peak signal-to-noise ratio and decreases the packet loss rate, the delay, and the energy consumption relative to the benchmark scheme, but the accuracy is only achieved up to 90%.

From the literature, it is suggested that video quality transmission is more important in the case of telemedicine and education sectors. Due to the bandwidth constraints, large volumes of data transmission and the mobility of the user vehicles, propagation path loss, etc., are reasons for the reduced video quality. The challenges for the transmission of quality video data are bandwidth limitation, compression and expansion of data speed, and latency in data transmission from the transmitter to the receiver. The existing H.264 and H.265 protocols in the 5G network video signal transmission give less accuracy and efficient data transmission. Suppose additional compression techniques are used at the transmitter and receiver devices. In that case, additionally, there is a big challenge to balance the quality of the video, the compression levels, and the latency in the video data display. To provide video compression without losing any video quality, the authors propose a novel concept of the LVT model with VIoT incorporation into the KNN merged H.265 protocol and H.264 protocols. The main objective of this proposed research is to improve video quality with less latency in the sensor network. The authors identified a significant improvement between the proposed KNN classifier-based H.265 protocol and the existing typical H.265 and H.264 protocols using performance metrics such as PSNR, Packet loss rate, SSIM, VMAF, latency, and throughput.

## 2. VIoT in Video Compression

One of the main challenges in the sensing phase is the development of VIoT-specific video data compression and LVT to the raw data shown in Figure 1.

The VIoT has already progressed with a set of global standards for MPEG and HEVC. Due to fundamental differences in video streams, adopting video compression in VIoT is an open research issue. In general, while compressing entertainment films, the high-definition reproduction of virtually all images is essential for comfortably watching video content. The ultimate goal of VIoT applications is to preserve the relevant information rather than the pixel value. As a result, keeping the spatial contents and settings of VIoT data is the essential quality for VIoT video compression and sensor data processing. Novel video compression algorithms fundamentally differ from current MPEG and HEVC standards that should maintain the best interpretation and background of the sensor data throughout the compression process. Recently, the development of a new video coding standard for machine communications produced enormous amounts of information for various applications. Because of the unique features of visual sensor data, VIoT systems can reveal more insights than those typical of IoT systems. These unique features enable VIoT systems to reach a wide range of new application industries by adding new aspects to existing IoT applications. The obtained visual sensor data can be pooled, analyzed, and interpreted using modern data modeling, machine learning, and supervised learning techniques. The resulting knowledge, which includes the recognition of behaviors and trends, reveals new perspectives that can affect each aspect of our lives, from better congestion control to crime prevention and from primary prevention to environmental protection [11,12,13].

### 2.1. LVT Model

Lossy Video Transmission (LVT) simulator is a framework for analyzing the impact of network congestion on the segmentation of video frames (at WVSNs) (on the decoder side). The main goal of WSNs in system simulators is scalability (i.e., their ability to grow) capability to manage large groups of nodes. Because LVT focuses solely on image quality assessments, it can work enormous sets of simulations with multiple images, procedures, and loss patterns. Simply put, LVT simulation consists of five major components or stages [14,15].

(1) Forward Feature Extraction: Using an image processing algorithm on an input image. This method yields a customized rendition of the original image [16].

(2) Packetization: The use of a packetization system. This step connects processed image data to packets.
(1)PSNRdB=10×log102(n−1)/2MSE

(3) Packet loss Modeling: This stage simulates data loss during network transmission. The losses are created randomly or by inputting a loss pattern file.

(4) Depacketization: The packetization scheme’s inverse application.

(5) Reverse Image Acquisition: The process of running the inverse image processing method. As a result, a version of the source image is created with some lost sequences.

(6) Error Hiding: An error concealment approach might be used to fill the gaps.

(A) LVT simulated model

(i) Proposed simulation model: The models used are fundamental. An incoming video frame *I* is an *(L, B)* matrix, *I = {Ir,c},* with *r*, *c* ϵ *Χ* ˄ 0 ≤ *r* ˂ *L* ˄ 0 ≤ *c* ˂ *B*; each *Ir,c* pixel has its *b* bits in each pixel, where *b* ϵ *R*+ considers the communication system *Ґ* by transmitting *I* in ⌊(*L* × *B* × *b*)/*m*⌋ with packets *P*, and where m states the number of bits allocated for video data transfer in a packet. During communication, each packet *pl* has a chance of being lost. Various loss models can be employed to accomplish this. Because packet drops are expected to occur over a wireless channel with one or more intermediary nodes, the path characteristics should be irrelevant to the simulation (clearly, the number of nodes and customized communication protocols may alter the loss rate). Averaging the well-received neighboring pixels yield an estimate of lost data for error concealing.

(ii) Encoding of Frames: As previously noted, the earliest form of LVT included a frame coding error-resilient computation technique. In a typical single block-based transmission, the image is first divided into $\frac{L\;X\;B}{L_{f}\;XB_{f}}$ blocks *F_i,j_*, where *L_f_* and *B_f_* are the length and breadth of the block frame, and *F_i,j_ = {I_rf_,c_f_}*, where, $L_{f}\;\leq \;r_{f}\;<\;L_{f}(i+1B_{f}\;\leq$ $c_{f}\;<\;B_{f}$(*j* + 1).

We allocate and deliver the *i*th block to the packet in a sequential transmission, with $t=\lceil \frac{i.b}{m}\rceil$. This sequence is disrupted by interleaving. It is a bijective function: V: *I*$\to \bar{I}$*,* where $\bar{I}$ is defined as a new bitmap where all original blocks *F_i,j_* can be kept in position ($\bar{I}\;\bar{J}$). An improved model covers sequential processes on a low-resource network (requiring less memory and calculations). During the packetization process, semi-pixel intensities are produced, but interleaving methods are utilized to select the data to put into the under-construction packets.

(B) Video Quality Assessment

LVT aims to support the measurement of the quality of produced image frames in WVSNs. Both subjective and objective assessment indicators for quality evaluation are conducted. Direct visualization of the rebuilt frames provides subjective judgment.

(i) Peak Signal-to-Noise Ratio

As shown below, the peak signal-to-noise ratio (PSNR) is employed to evaluate the restoration quality of the suggested image restoration by SR, wherein MSE denotes the mean squared error.

(ii) Mean absolute deviation (MAD)

The mean absolute deviation IQA model is formulated as a deviation of spatial regions from its average values in Equations (2) and (3):(2)$$MAD\_measure=\frac{MAD\left(input\right)}{MAD\left(output\right)}$$
(3)$$MAD=\overset{n}{\underset{i-1}{\sum }}\frac{\left|x_{i}-\bar{x}\right|}{n}$$
where *x_i_* = pixel values; $\bar{x}$ = Mean value; *n* = number of pixels value;

(iii) Structural similarity index (SSIM)

The structural similarity index (SSIM) is used to compare the resemblance of low input resolution and high-resolution images using orthogonal quantitative measures such as luminance (*µ*) and contrast (*σ*) as follows and as shown in Equations (4) and (5)
(4)$$C_{L}\left(I\cdot I_{0}\right)=\frac{2\mu I\mu I_{0}+C1}{\mu I^{2}+\mu I_{0}^{2}+C1}$$
(5)$$C_{c}\left(I\cdot I_{0}\right)=\frac{2\sigma I\sigma I_{0}+C2}{\sigma I^{2}+\sigma I_{0}^{2}+C2}$$
where *C*1 and *C*2 are constants, the picture structure is determined by normalizing, as illustrated in Equation (6):(6)$$S=\left(I-\mu I\;\right)/\sigma I$$

The measure of the structural similarity is evaluated based on its correlations.

### 2.2. Encoding Standard

#### 2.2.1. Video Coding Layer

The encoding standard used is H.265. It consists of a video coding layer that encodes and transmits the video over the network. The video is broken up into three frames. *I* is the primary/reference frame that is unaffected by other frames. The P frame is interdependent on its previous one but can be decoded independently. A B frame can be used to refer to another B frame. All the frames in sequence can be defined as a Group of Pictures (GoP). The concept of Macroblocks is used in H.265. The typical size of each macroblock can be different; 16 × 16 or 8 × 8 luma/chroma channels define the best size. Contemporary to macroblock is Coding Tree Unit; both can be combined to form a slice. The slice can be decoded separately even if the same frame is unavailable [17,18,19,20].

#### 2.2.2. Network Abstraction Layer

The generated encoded video from the VCL is converted into bits by NAL, which further improves the efficiency of the video transmission. For example, RTP can be converted into mp3 or mp4 for video storage, and NAL can be classified into two types—VCL NAL and Non-VCL NAL units. The former carries the data for the video requirements, and the latter has additional overhead information. H.265 defines different types of NAL standards [21,22].

## 3. Methodology

The authors applied the proposed LVT model with VIoT sensors to the modified H.265 protocol. The above figure shows the machine learning-based H.265 protocol containing the KNN Classifier for the feature extractions by training and testing the samples. In general, H.265 protocol has video compression and extractors, but it contains low computational speed. Due to this, the latency will be increased, which causes the efficiency decrement. To improve the latency by reducing the computational rate, the authors proposed merging the KNN classifier in the H.265 protocol, as shown in Figure 2.

The LVT and VIoT sensors are placed at the transmitter and receiver to process the video. The KNN-based H.265 protocol is applied to compress and extract faster in the network devices. The video transmission takes care of the LVT using compression and extraction, and the VIoT sensors are for measuring the video quality. These two are computed based on the KNN-H.265 protocol. The main objective is to balance the process between the LVT and VIoT without any latency, as well as to provide excellent video quality.

A simple architecture was developed with three modules: a transmitter, a packet failure model, and a receiver. The proposed architecture comprises compressed video data using a modified H.265 protocol to realize the packet loss and improve video streaming [23]. The compressed video data is transmitted using the 802.11p protocol. The streamed data utilizing the virtual network interface is transmitted to the receiver. Finally, the video is stored in RTP packets as a distorted video. The evaluation follows considerations and different encoding parameters [24,25,26]. As a preliminary example, we investigated the performance evaluation of new network coding techniques. One proposed evaluation methodology was streaming videos over an emulated network with a packet delivery ratio even without network coding. Then, once finished with network coding, we compared the QoE of both by calculating the quantitative QoE metrics. The authors’ second method was where packets were transferred to the network with network coding [27].

### 3.1. Transmitter

The KNN-based H.265 model was used as a reference for the encoder at the transmitter section. Video data such as live streaming or any recorded video can be transmitted from base station transmitter to user receiver device. The primary parameters used while transmitting the video are listed in Table 1. Group of Pictures: Generally, any GoP structure for the evaluation of packet loss can be used, but the structure used here is I-B-B-B-P-B-B-B-I, with a GoP size of 8.

### 3.2. Packet Failure Module

To determine the errors in the packet sensing and transmission, four types of packet errors—propagation, sensing, busy receiver, and collision errors—were analyzed in [28,29,30,31], which helped us determine the improvement of throughput efficiency. This model used a hidden Markov model to calculate interference and propagation losses.

### 3.3. Receiver

The main objective of this design is to ensure data transfer without packet delays. The decoding of the received data should be conducted one at a time in the receiver section so as to avoid the impact on QoE metrics. The second criterion is that the loss sequence should match the parameter set, e.g., if there is a 10% increase in the PLR, then the same 10% increase must also be seen with packet loss. This requirement is satisfied using a random number generator of the packet loss model with a number of seeds fixed. Next, by keeping payloads fixed, RTP packets are transmitted by H.265, and the loss sequence should not depend on cross-traffic from any other application [28,29,30,31,32,33].

## 4. Results

This section discusses results concerning the packet loss environment and the delay in investigating the behavior of prospective metrics in the presence of packet delay in various IoT scenarios. The proposed protocol was assessed and compared with existing state-of-the-art works using MATLAB software. The assumptions for the simulation environment are shown in Table 2.

Figure 3 represents the relationship between frame number and PSNR. When the frame number increases, the PSNR level decreases. If the PSNR is decreased, the image’s intensity will decrease. Simulated data show that a PSNR value of 30 dB at frame 300 is decreased to 15 dB when the frame number is increased from 300 to 600 for the existing H.264 protocol. In contrast, the existing H.265 protocol’s PSNR value of 34 dB at frame 300 is decreased to 25 dB when the frame number is increased from 300 to 600. In the existing two protocols, there is no ML technique used. However, in the proposed model of H.265, a KNN classifier is used. Due to this, after compression, the video PSNR value is 40 dB at frame number 300 and 37 dB at frame number 600, as shown in Table 3.

Figure 4 represents the relationship between frame number and SSIM. When the frame number increases, the SSIM level decreases. If the SSIM is decreased, the image’s quality will decrease. The existing H.264 protocol SSIM value is 30 dB at frame number 300 and is decreased to 11 dB when the frame number is increased from 300 to 600. The existing H.265 protocol SSIM value is 33 dB at frame number 300 and is decreased to 29 dB when the frame number is increased from 300 to 600. In the existing two protocols, there is no ML technique used. However, in the proposed model of H.265, a KNN classifier is used. Due to this, the video quality value is 40 dB at frame number 300 and 29 dB at frame number 600 after compression.

Figure 5 represents the relationship between packet loss ratio and throughput. Here, the authors estimated the throughput in Mbps because the 5G and IoT networks require a minimum Mbps speed. When the PLR increases, the throughput also varies. If the throughput decreases, the data delivery fails and will increase. The existing H.264 protocol throughput ranges from 450 Mbps to 460 Mbps when the PLR ranges from 0 to 12%. The existing H.265 protocol throughput varies from 800 Mbps to 810 Mbps when the PLR ranges from 0 to 12%. The proposed KNN-H.265 throughput varies from 1100 Mbps to 1110 Mbps when the PLR ranges from 0 to 12%, as shown in Table 4. Hence, the proposed method gives higher throughput performance when compared with the existing methods.

Figure 6 represents the relationship between throughput and delay. Delay is an essential parameter in data transmission, especially for video signal transmission. The parameter delay depends on channel parameters such as propagation environment, reflection, distance, etc. When the delay is increased from 0 to 600 ms, the existing H.265 gives better throughput than the H.264 protocol. When the existing two protocols are compared with the proposed KNN-H.265 protocol, the proposed protocol provides better throughput. At 300 m s of delay, the proposed method improves by 400 Mbps and 210 Mbps compared to the H.264 and H.265 protocols, respectively, as shown in Table 5.

Figure 7 represents the relationship between distance and delay. Delay is an essential parameter in data transmission and depends on the distance between the transmitter and the receiver. The parameter delay depends on channel parameters such as propagation environment, reflection, distance, etc. When the distance increases from 0 to 1000 m, the existing H.265 gives less delay than the H.264 protocol. When the existing two protocols are compared with the proposed KNN-H.265 protocol, the proposed protocol provides less delay/latency. At a 500 m distance, the proposed method has a 410 m s and 190 m s lower delay when compared with the H.264 and H.265 protocols, respectively, and is shown in Table 6.

We employed a learning curve to analyze the KNN model with the H.265 protocol and with LVT model detection. The learning curve of the KNN model shows that the training and test converges together, as we observe in Figure 8. Thus, we conclude that the performance of the KNN model tends to improve substantially a with larger number of training samples. Moreover, the learning curve for KNN, demonstrated in Figure 9, shows that the model is not suffering from overfitting or under fitting, as both the training and test accuracy tends to increase with an increase in the number of observations.

The authors of the research in [28] proposed a single buffer instead of a double buffer with Multi-Sensory (MS) alerts and estimated the performance of the MS-KNN classifier-based single buffer with the double buffer system. The performance metrics were used to estimate are sensing error, collision error and propagation error, etc. In this proposed research paper, the authors applied the LVT technique to the KNN classifier-enabled H.265 protocol for the VIoT. The performance of the proposed method purely depends on the video signal transmission in the channel. So, the authors estimated parameters such as throughput, delay, packet loss ratio, frame number and PSNR etc. They also estimated the performance of the proposed LVT-KNN classifier with the mentioned parameters in [28], such as sensing error, propagation error and collision error. The performance of the proposed method was compared with the existing MS-KNN single buffer [28], and the higher-performing method was identified and is shown in Figure 9, Figure 10 and Figure 11.

Figure 9 represents the relationship between distance and probability of collision error. The distance between the transmitter and the receiver increases, and the collision error also varies. At a 200 m distance, the number of users is higher, and a huge number of video data packets transmission is required. Due to this huge volume of data transmission, collision errors occurred. So, there is a higher probability of getting collision error when at a long distance. At this distance, the MS-KNN single buffer (existing) gives a 16% higher collision error probability than the proposed LVT-KNN classifier method.

Figure 10 represents the relationship between distance and probability of propagation error. The distance between the transmitter and the receiver increases, and the propagation error also increases. Because the distance is increased, the signal strength is going to decrease, which causes an increase in propagation errors. At a 400 m distance, the MS-KNN single buffer (existing) gives a 5% higher propagation error probability than the proposed LVT-KNN classifier method.

Figure 11 represents the relationship between distance and probability of sensing error. The distance between the transmitter and the receiver increases, and the sensing error also increases. At a 400 m distance, the MS-KNN single buffer (existing) gives a 15% higher sensing error probability than the proposed LVT-KNN classifier method.

## 5. Conclusions

The new Internet of Video Things refers to an emerging type of IoT system incorporating wireless visual sensors at the front end (VoIT). The H.265 coder with a LVT programmable framework for simulating video communications in Wireless Visual Sensor Networks has been experimented on with error-tolerant methodologies such as block connecting. The KNN classifier-based machine learning techniques with LVT and VoIT were applied to the H.265 protocol for the fast computation and transmission of video data packets from transmitter to receiver. In this context, the proposed protocol’s performance was compared and analyzed with the existing H.264 and H.265 protocols. The parameters of throughput, delay, PSNR, Packet loss ratio, and SSIM were used to estimate the protocol’s performance. The proposed protocol achieves improved performance by reducing latency by up to 190 ms compared to the existing state-of-the-art H.264 and H.265 protocols; in terms of throughput, the proposed protocol achieves improved capacity compared with the H.264 and H.265 protocols, with a nominal 300 ms of delay. Therefore, the performance of the proposed protocol can achieve better results when compared with the existing protocols with regard to lower latency and higher throughput. The proposed work was compared with the existing MS-KNN single buffer model. The proposed work gives 16%, 5% and 15% better improvement in terms of collision error, propagation error and sensing error, respectively. There is a plan to extend this work in the future by applying LSTM-based machine learning algorithms using the KNN-H.265 protocol.

## Figures and Tables

**Figure 1 sensors-23-05072-f001:**
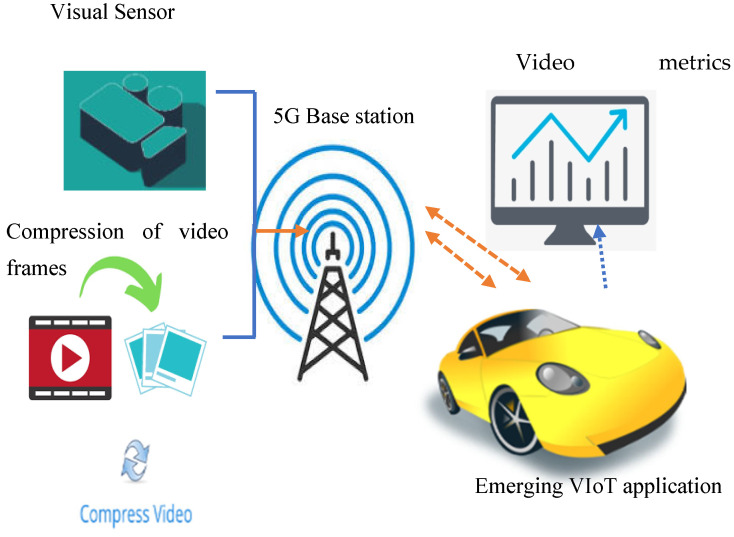
Illustration of VIoT applications.

**Figure 2 sensors-23-05072-f002:**
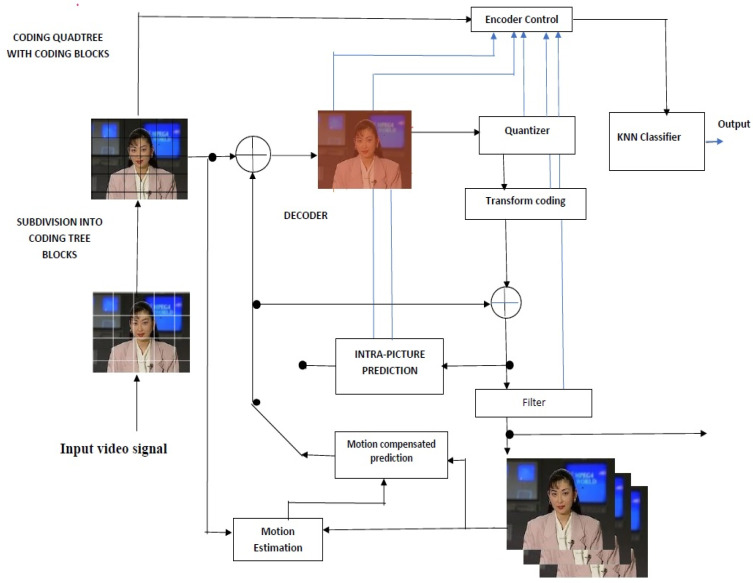
KNN classifier-based H.265 protocol.

**Figure 3 sensors-23-05072-f003:**
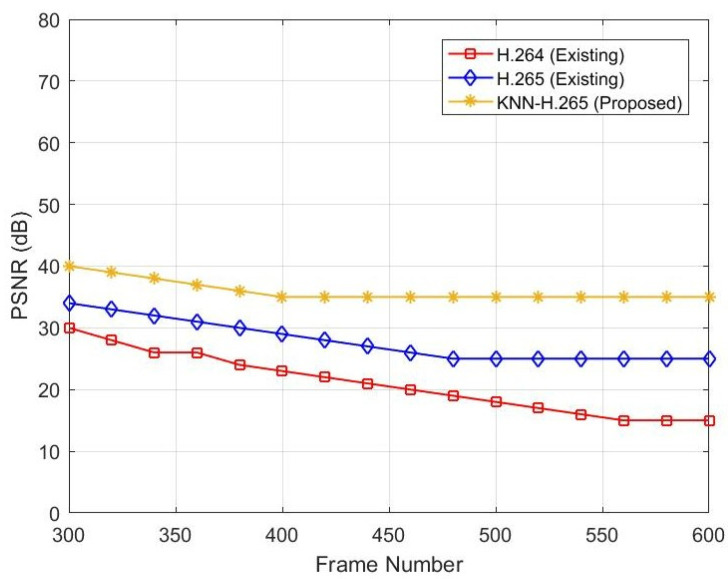
PSNR vs. frame number.

**Figure 4 sensors-23-05072-f004:**
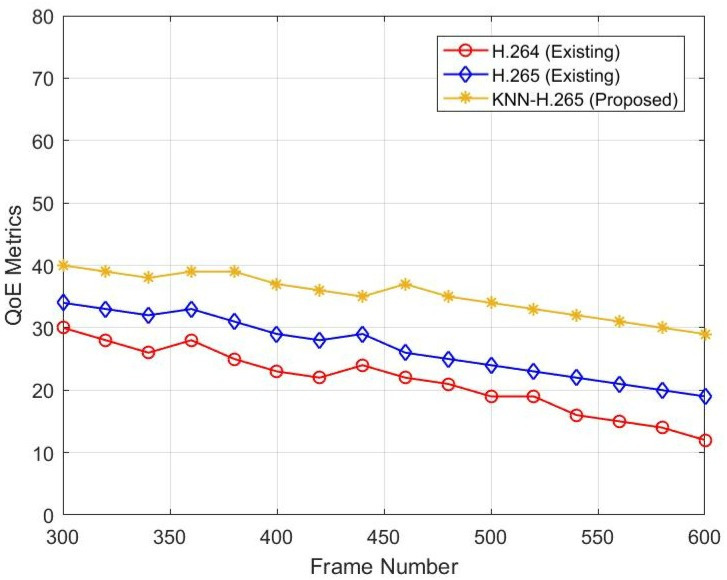
SSIM versus frame numbers.

**Figure 5 sensors-23-05072-f005:**
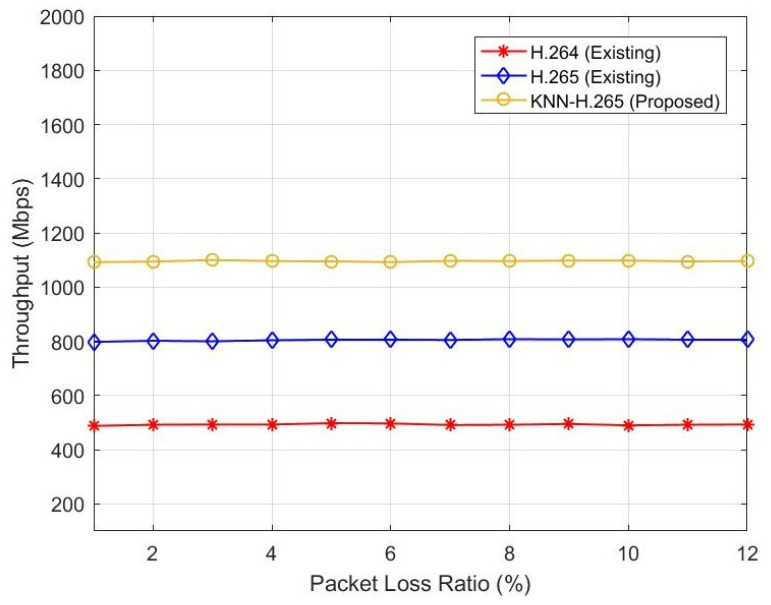
Throughput vs. packet loss ratio.

**Figure 6 sensors-23-05072-f006:**
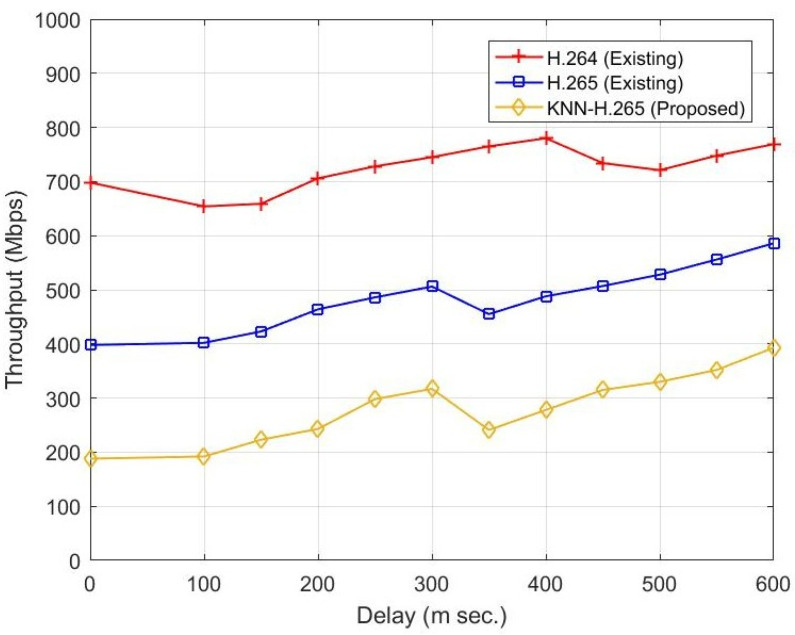
Throughput vs. delay.

**Figure 7 sensors-23-05072-f007:**
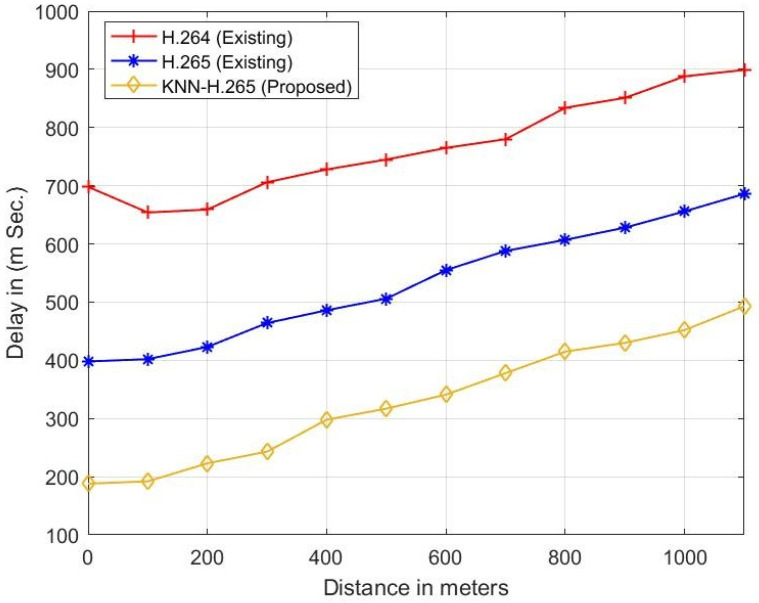
Distance vs. delay.

**Figure 8 sensors-23-05072-f008:**
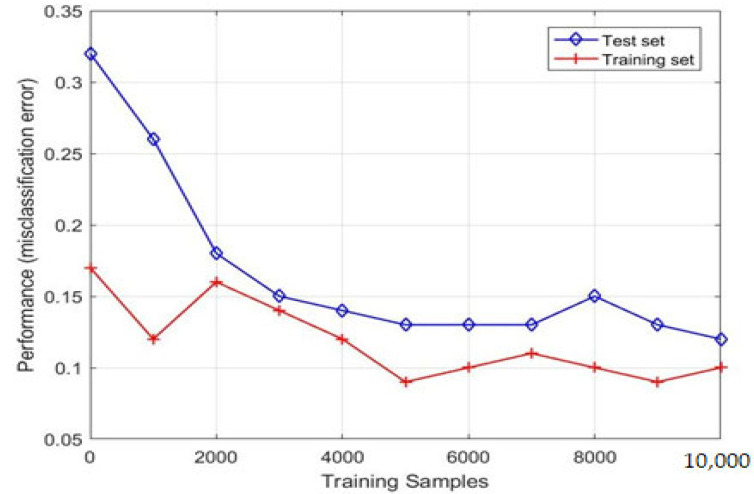
Learning curve for KNN model.

**Figure 9 sensors-23-05072-f009:**
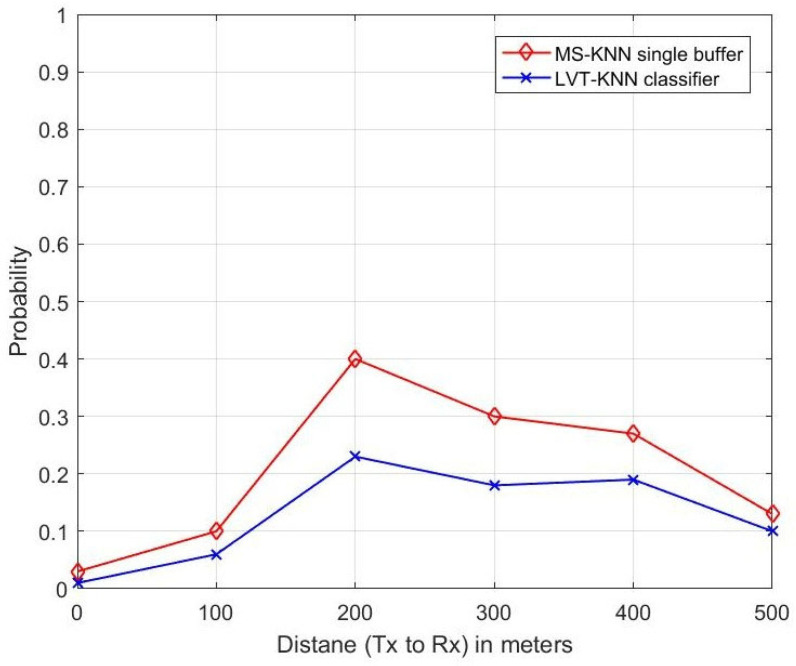
Collision Error.

**Figure 10 sensors-23-05072-f010:**
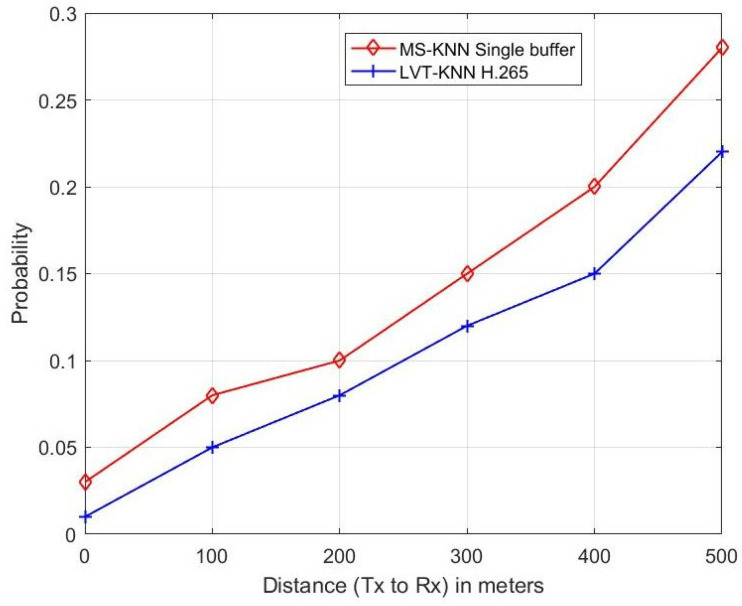
Propagation Error.

**Figure 11 sensors-23-05072-f011:**
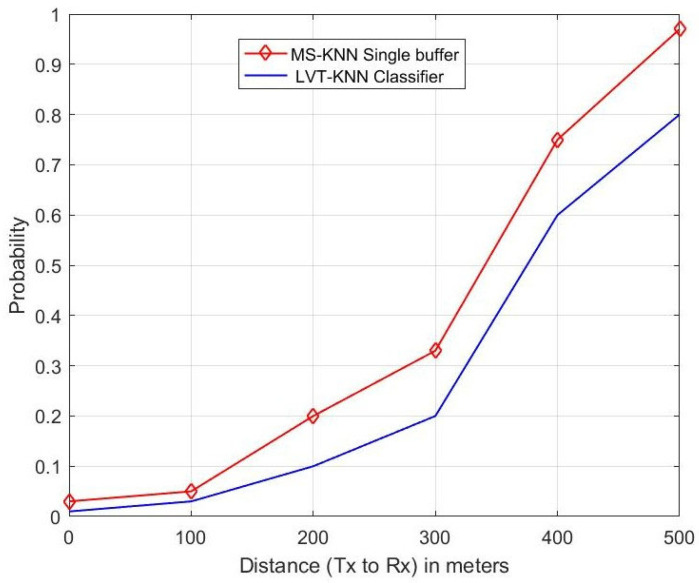
Sensing Error.

**Table 2 sensors-23-05072-t002:** List of Encoding Parameters.

Parameter	Value
Video User Profile	Video Data of the user
Compressor, Transmission	Compressor, Transmission
Simulation Software	FFMPEG version 4.2.2
VMAF	Version 1.8.1
Quality	480⇥1080
Frame Rate	Frame frequency of the compressed video
Video data rate	64M bit/s
Strategy for error coverage	None
libx265 Encoder Parameter	
Compressor Profile	Main
Compressor Level	4
Group of Pictures Structure	I-B-B-P-B-B-P-I
Packet Measurement Mode	binary NAL element
Scalable B frame Mode	Deactivated
JCTVC HEVC Reference Encoder Parameter	
Compressor Profile	Main
GOP Structure	I-B-B-B-P-B-B-B-I
Scalable B frame Mode	Active

**Table 3 sensors-23-05072-t003:** Comparative performance between frame number and PSNR.

Frame Number	PSNR (dB)		
	H.264	H.265	Proposed
300	30	32	40
350	25	31	37
400	22	29	35
450	20	37	33
500	18	25	33
550	13	25	33
600	13	25	33

**Table 4 sensors-23-05072-t004:** Comparative performance between PLR vs. Throughput.

PLR	Throughput (Mbps)		
	H.264	H.265	Proposed
2	492	802	1094
4	493	804	1096
6	497	806	1092
8	492	808	1096
10	490	808	1098
12	493	806	1096

**Table 5 sensors-23-05072-t005:** Comparative performance between delay and Throughput.

Frame Number	Throughput (Mbps)		
	H.264	H.265	Proposed
0	698	398	188
100	659	423	223
200	728	464	298
300	765	506	241
400	734	488	315
500	721	528	352
600	769	586	393

**Table 6 sensors-23-05072-t006:** Comparative performance between distance and delay.

Distance (m)	Delay (m s)		
	H.264	H.265	Proposed
0	698	388	180
200	649	413	213
400	718	476	288
600	755	545	331
800	834	601	415
1000	888	656	452

## Data Availability

Not applicable.

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
