# Peer review of "Optimized Visual Internet of Things for Video Streaming Enhancement in 5G Sensor Network Devices"

_sensors, 2023, doi:10.3390/s23115072_

Round 1
Reviewer 1 Report
The paper mentioned originality and stated that it improves the video streaming quality by using the proposed framework Lossy Video Transmission (LVT) for simulating the effect of network congestion on the performance of encrypted static images sent over wireless sensor networks. However, simulating the effect of network congestion can not improve only if it is used to activate a system or solution.
The introduction did not specify clearly, the research problem, research question and research objectives.
The paper needs related work that also needs critical review analysis.
There is no comparison with other works to show the contribution and specify clearly the originality.
The paper needs to show the latency before and after the enhancement.
Author Response
The paper mentioned originality and stated that it improves the video streaming quality by using the proposed framework Lossy Video Transmission (LVT) for simulating the effect of network congestion on the performance of encrypted static images sent over wireless sensor networks. However, simulating the effect of network congestion can not improve only if it is used to activate a system or solution.
- The introduction did not specify clearly, the research problem, research question and research objectives.
Response: As per the reviewer’s comments, the introduction section is revised, research gaps are mentioned and the research problem is also stated in the revised manuscript.
|
Author’s names |
Methods used |
limitations |
|
Chen C.W. et al. |
Large-scale visual sensor-based Internet of Video Things, Dynamic Bandwidth and Wavelength Allocation (DWBA) scheme |
Due to Bandwidth constraints, the packet loss is more, and the transmission delay is more |
|
Ganesan, E. et al. |
5G-enabled adaptive, scalable video streaming multicast in a software-defined NG-EPON network |
Quality of service parameters performance is poor |
|
Ni, CT et al. |
HEVC/H.265, H.264/AVC are used for video compression |
50% of video quality is lost because of H.264, and 45.33% of Video Quality is lost because of H.265 |
|
Om, K. et al. |
Accuracy is estimated by applying Long Short Term Memory, Convolutional Neural Networks and Sequence-to-Sequence models to H.264 and H.265 |
Poor accuracy is obtained because of the conventional neural networks |
|
Zaki, F et al. |
CtuNet framework-based H.264, and H.265 protocols are used |
Because of the low data rate, the transmission delay is increased more |
|
Chen, L et al. |
Protocol identification (PI) and video quality enhancement (VQE) tasks on IIoT edge devices using deep neural networks (DNNs) |
97% of accuracy is achieved, but latency is increases |
|
Xiao, Y et al. |
The reinforcement learning algorithm is used for video streaming in IoT |
Increases the peak signal-to-noise ratio and decreases the packet loss rate, the delay, and the energy consumption relative to the benchmark scheme, but the accuracy is achieved up to 90% only. |
- The paper needs related work that also needs critical review analysis.
Response: As per the reviewer’s suggestion, a review analysis is done and incorporated into the revised manuscript.
From the literature, it is motivated that video quality transmission is more important in the case of telemedicine and education sectors. Due to the bandwidth constraints, large volumes of data transmission and the mobility of the user vehicles, propagation path-loss, etc., are reasons for the reducing the video quality. The challenges for transmission of quality video data transmission are bandwidth limitation, compression and expansion of data speed, and latency in data transmission from the transmitter to receiver. The existing H.264 & H.265 protocols in the 5G network video signal transmission give less accuracy and efficient data transmission. Suppose additional compression techniques are used at the transmitter and receiver devices. In that case, additionally, there is a big challenge to balance the quality of the video, the compression levels, and the latency in the video data display. To provide video compression without losing any video quality, the authors proposed a novel concept of the LVT model with VIoT incorporation in the KNN merged H.265 protocol and H.264 protocols. The main objective of this proposed research work is to improve video quality with less latency in the sensor network. A significant improvement was identified by the authors in between proposed KNN classifier-based H.265 protocol and existing typical H.265 & H.264 protocols using the performance metrics such as: PSNR, Packet loss rate, SSIM, VMAF, latency, and throughput.
- There is no comparison with other works to show the contribution and specify clearly the originality.
Response: The author’s contributions are
- Applied the KNN classifier to the H.265 Protocol and further VIoT enabled LVT models are emerged in it.
- 2. The comparison results are developed by compare with the proposed KNN-H.265 protocol with normal H.264 & H.265 Protocols and also identified the better performance with respect to the parameters of PSNR, SSIM, throughput and latency. The comparison results are shown in the revised manuscript.
- The paper needs to show the latency before and after the enhancement.
Response: As per the reviewer’s suggestion, the latency is estimated before and after the enhancement.
Figure 7 Distance vs Delay
Figure 7 represents the relationship between distance versus delay. Delay is an essential parameter in data transmission and depends on the distance between the transmitter and the receiver. The parameter delay depends on the channel parameters like propagation environment, reflection, distance, etc. When the distance increases from 0 to 1000 meters, the existing H.265 gives less delay than the H.264 protocol. The existing two protocols are compared with the proposed KNN-H.265 protocol, the proposed protocol provides less delay/latency. At 500 meters of distance, the proposed method gives 410 m sec and 190 m sec less delay when compared with the H.264 and H.265 protocols, respectively.
Hanging sentence; who identified the significant improvement or for whom? This sentence compares what ?
Please complete …

Reviewer 2 Report
1) The quality of the Figures is very low. Even, I can't read Figure 1.
2) All Equations must be re-presented, they must be written by Equation software. Even, Equation (1) is missing/error.
It's noted that all Equations need to be numbered.
3) What simulation software did the authors use? More analyses should be presented.
4) The relationship between the research problem and Machine Learning should be highlighted.
In the reviewer's opinion, the authors should perform double-checking throughout the manuscript. I felt this is a draft.
Author Response
Response to reviewer comments
- The quality of the Figures is very low. Even, I can't read Figure 1.
Response: As per the reviewer comment, Figure 1 quality is enhanced.
- All Equations must be re-presented, they must be written by Equation software. Even, Equation (1) is missing/error.
Response: As per the reviewer’s suggestions, Equations are typed in the editable form in the revised manuscript.
- It's noted that all Equations need to be numbered.
Response: As per the reviewer’s suggestion, numbering is given to all equations
- What simulation software did the authors use? More analyses should be presented.
Response: The results are plotted using MATLAB software and the analyses are improved in the revised manuscript.
- The relationship between the research problem and Machine Learning should be highlighted.
Response: The authors applied the proposed LVT model with VIoT sensors to the modified H.265 protocol. The above figure shows the machine learning-based H.265 protocol containing KNN Classifier for the feature extractions by training and testing the samples. In general, H.265 protocol has video compression and extractors, but it contains low computational speed. Due to this, the latency will be increased, which causes the efficiency decrement. To improve the latency by reducing the computational rate, the authors proposed merging the KNN classifier in the H.265 protocol, shown in below Figure.
The LVT and VIoT sensors are placed at the transmitter and receiver to process the video. The KNN-based H.265 protocol is applied to compress and extract faster in the network devices. The video transmission takes care of the LVT with compression and extraction, and the VIoT sensors for measuring the video quality. These two are computed based on the KNN-H.265 protocol. The main objective is to balance the process between the LVT and VIoT without any latency, as well as excellent video quality.
- In the reviewer's opinion, the authors should perform double-checking throughout the manuscript. I felt this is a draft.
Response: As per the reviewer suggestion’s, the paper is verified and the English grammar is also corrected by a native English speaker.

Reviewer 3 Report
1. Authors should emphasize contribution and novelty in the abstract. Moreover, the introduction needs to clarify the (1) motivation, (2) challenges, (3) contribution, (4) objectives, and (5) significance/implication.
2. It helps to appreciate the paper by having a related work section. The authors should consider more recent research done in the field of their study (especially in the years 2021 and 2022 onwards). The reader may want to see how this work differs from other previous works.
3. The authors should clearly describe related work in more detail, contrasting the limitations of the related works. Moreover, the reviewer recommend to ease the overview related works by using overview tables.
4. There is no discussion of user requirements, technological options and support for the decisions made at the design. The authors should include more technical details and explanations.
5. This paper should summarize those results and give a comprehensive performance comparison with previous works (Optimized). It needs to highlight the research main contribution with some brief indications and numerical improvement percentages in section of result.
6. The article missed presenting the research novelty (Machine Learning). In the sense that they do not highlight what is missing from each of the other proposals. The authors should provide enough proof to convince the reader of superiority of the proposed schemes over the existing works.
7. The conclusion and future work part can be extended to have a better understanding of the approach and issues related to that which can be taken into consideration for future work.
Author Response
Response to Reviewer Comments
- Authors should emphasize contribution and novelty in the abstract. Moreover, the introduction needs to clarify the (1) motivation, (2) challenges, (3) contribution, (4) objectives, and (5) significance/implication.
Response: As per the reviewer’s suggestion, the abstract and introduction sections are updated in the revised manuscript.
From the literature, it is motivated that video quality transmission is more important in the case of telemedicine and education sectors. Due to the bandwidth constraints, large volumes of data transmission and the mobility of the user vehicles, propagation path-loss, etc., are reasons for the reducing the video quality. The challenges for transmission of quality video data transmission are bandwidth limitation, compression and expansion of data speed, and latency in data transmission from the transmitter to receiver. The existing H.264 & H.265 protocols in the 5G network video signal transmission give less accuracy and efficient data transmission. Suppose additional compression techniques are used at the transmitter and receiver devices. In that case, additionally, there is a big challenge to balance the quality of the video, the compression levels, and the latency in the video data display. To provide video compression without losing any video quality, the authors proposed a novel concept of the LVT model with VIoT incorporation in the KNN merged H.265 protocol and H.264 protocols. The main objective of this proposed research work is to improve video quality with less latency in the sensor network. A significant improvement was identified by the authors in between proposed KNN classifier based H.265 protocol and existing typical H.265 & H.264 protocols using the performance metrics such as: PSNR, Packet loss rate, SSIM, VMAF, latency, and throughput.
.
- It helps to appreciate the paper by having a related work section. The authors should consider more recent research done in the field of their study (especially in the years 2021 and 2022 onwards). The reader may want to see how this work differs from other previous works.
Response: As per the reviewer’s suggestion, the introduction section is updated with very recent years research papers in the revised manuscript.
- Chen, C. W. (2020). Internet of Video Things: Next-generation IoT with visual sensors. IEEE Internet of Things Journal, 7(8), 6676-6685.
- Ganesan, E., Hwang, I. S., & Liem, A. T. (2021, November). 5G-enabled multicast for scalable video streaming in software-defined NG-EPON. In 2021 International Symposium on Intelligent Signal Processing and Communication Systems (ISPACS) (pp. 1-2). IEEE.
- Ni, C. T., Huang, Y. C., & Chen, P. Y. (2023). A Hardware-Friendlyand High-Efficiency H. 265/HEVC Encoder for Visual Sensor Networks. Sensors, 23(5), 2625.
- Om, K., McGill, T., Dixon, M., Wong, K. W., & Koutsakis, P. (2022). H. 264 and H. 265 video traffic modeling using neural networks. Computer Communications, 184, 149-159.
- Zaki, F., Mohamed, A. E., & Sayed, S. G. (2021). CtuNet: a deep learning-based framework for fast CTU partitioning of H265/HEVC intra-coding. Ain Shams Engineering Journal, 12(2), 1859-1866.
- Chen, L., Liu, L., & Zhang, L. (2022). An efficient Industrial Internet of Things video data processing system for protocol identification and quality enhancement. IET Cyber‐Physical Systems: Theory & Applications.
- The authors should clearly describe related work in more detail, contrasting the limitations of the related works. Moreover, the reviewer recommend to ease the overview related works by using overview tables.
Response: As per the reviewer’s comment, the related work is placed in the table format.
|
Author’s names |
Methods used |
limitations |
|
Chen C.W. et.al |
Large scale visual sensor based Internet of Video Things, Dynamic Bandwidth and Wavelength Allocation (DWBA) scheme |
Due to Bandwidth constraint the packet loss is more and transmission delay is more |
|
Ganesan,E. et al |
5G-enabled adaptive scalable video streaming multicast in a software defined NG-EPON network |
QoS parameters performance is poor |
|
Ni, C.T. et al |
HEVC/H.265, H.264/AVC are used for video compression |
50% of video quality is lost because of H.264 and 45.33% of Video Quality loss because of H.265 |
|
Om,K. et al |
Accuracy is estimated by applying Long Short Term Memory, Convolutional Neural Networks and Sequence-to-Sequence models to H.264 and H.265 |
Poor accuracy is obtained because of the conventional neural networks |
|
Zaki,F et al |
CtuNet frame work based H.264 and H.265 protocols are used |
Because of low data rate the transmission delay is increased more |
|
Chen,L et al |
Protocol identification (PI) and video quality enhancement (VQE) tasks on IIoT edge devices using deep neural networks (DNNs) |
97% of accuracy is achieved but latency is increases |
|
Xiao,Y et al |
Reinforcement learning algorithm is used for video streaming in IoT |
Increases the peak signal-to-noise ratio and decrease the packet loss rate, the delay, and the energy consumption relative to the benchmark scheme, but the accuracy is achieved up to 90% only. |
- There is no discussion of user requirements, technological options and support for the decisions made at the design. The authors should include more technical details and explanations.
Response: As per the reviewer’s suggestion the technical details are mentioned in the revised manuscript.
The authors applied the proposed LVT model with VIoT sensors to the modified H.265 protocol. The above figure shows the machine learning-based H.265 protocol containing KNN Classifier for the feature extractions by training and testing the samples. In general, H.265 protocol has video compression and extractors, but it contains low computational speed. Due to this, the latency will be increased, which causes the efficiency decrement. To improve the latency by reducing the computational rate, the authors proposed merging the KNN classifier in the H.265 protocol, shown in Figure 2.
Figure 2. KNN classifier-based H.265 protocol
The LVT and VIoT sensors are placed at the transmitter and receiver to process the video. The KNN-based H.265 protocol is applied to compress and extract faster in the network devices. The video transmission takes care of the LVT with compression and extraction, and the VIoT sensors for measuring the video quality. These two are computed based on the KNN-H.265 protocol. The main objective is to balance the process between the LVT and VIoT without any latency, as well as excellent video quality.
3.1. Transmitter
libx265 based H.265 model used as reference for the encoder. The primary parameters used while sending the video is listed in Table 1.Group of Pictures: In general any GoP structure for the evaluation of packet loss can be used but the structure used here is I-B-B-B-P-B-B-B-I with GoP size of 8.
3.2. Packet failure module
To determine the probability of packets sensed and transmitted, errors need to be taken into account which occurs during packet transmission. The four types of packet errors Propagation error, Sensing error, Busy Receiver error and Collision error were analyzed in [19-22] helped us to determine the improvement of throughput efficiency. This model used Hidden Markov Model to calculate interference and propagation losses.
3.3. Receiver
The major goal of this design is to ensure that data is sent without packet delays. In order to achieve high level of accuracy several constraints are analyzed. The encoding is done once at a time to avoid the impact on QoE metrics, which reduces the necessity of encoding several times when multiple streams are needed. The second criterion is the loss sequence should match the parameter set e.g. If there is 10% increase in the PLR then same 10% increase must be seen with packet loss also. This requirement is satisfied by using random number generator of the packet loss model with number of seeds fixed. Next by keeping payloads fixed, RTP packets are transmitted by H.265, the loss sequence should not depend on cross traffic from any other application.
Table 2. List of Encoding Parameters
|
Parameter |
Value |
|
Video User Profile |
Video Data of the user |
|
Compressor, Transmission |
Compressor, Transmission |
|
Simulation Software |
FFMPEG version 4.2.2 |
|
VMAF |
Version 1.8.1 |
|
Quality |
480⇥ 1080 |
|
Frame Rate |
Frame frequency of the compressed video |
|
Video data rate |
64M bit/s |
|
Strategy for error coverage |
None |
|
libx265 Encoder Parameter |
|
|
Compressor Profile |
Main |
|
Compressor Level |
4 |
|
Group of Pictures Structure |
I-B-B-P-B-B-P-I |
|
Packet Measurement Mode |
binary NAL element |
|
Scalable B frame Mode |
Deactivated |
|
JCTVC HEVC Reference Encoder Parameter |
|
|
Compressor Profile |
Main |
|
GOP Structure |
I-B-B-B-P-B-B-B-I |
|
Scalable B frame Mode |
Active |
- This paper should summarize those results and give a comprehensive performance comparison with previous works (Optimized). It needs to highlight the research main contribution with some brief indications and numerical improvement percentages in section of result.
Response: As per the reviewer’s suggestion, all the variables improvement is represented in the results.
Figure 3. PSNR vs. frame number
Figure 3 represents the relation between frame number versus PSNR. When the frame number increases, the PSNR level decreases. If the PSNR is decreased, the image's intensity will decrease. Simulated data interpret that if the PSNR value of 30 dB at frame 300 is decreased to 15 dB when the frame number is increased from 300 to 600 for the existing H.264 protocol. In contrast, the existing H.265 protocol's PSNR value of 34 dB at frame 300 is decreased to 25 dB when the frame number is increased from 300 to 600. In the existing two protocols, there is no ML technique used. But in the proposed model of H.265, a KNN classifier is used. Due to this, after compression, the video PSNR value is 40 dB at 300 frame number and 37 dB at 600 Frame number.
Figure 4. SSIM versus frame numbers
Figure 4 represents the relation between frame number versus SSIM. When the frame number increases, the SSIM level decreases. If the SSIM is decreased, the image's quality will decrease. The existing H.264 protocol SSIM value is 30 dB at frame number 300 and is decreased to 11 dB when the frame number is increased from 300 to 600. The existing H.265 protocol SSIM value is 33 dB at frame number 300 and is decreased to 29 dB when the frame number is increased from 300 to 600. In the existing two protocols, there is no ML technique used. But in the proposed model of H.265, a KNN classifier is used. Due to this, the video quality value is 40 dB at 300 frame number and 29 dB at 600 Frame number after compression.
Figure 5 Throughput vs. packet loss ratio
Figure 5 represents the relation between packet loss ratio versus throughput. Here the authors estimated the throughput in Mbps because the 5G and IoT networks require minimum Mbps speed. When the PLR increases, the throughput also varies. If the throughput decreases, the data delivery is failed and will increase. The existing H.264 protocol throughput ranges from 450 Mbps to 460 Mbps when the PLR ranges from 0 to 12%. The existing H.265 protocol throughput is varied from 800 Mbps to 810 Mbps when the PLR ranges from 0 to 12%. The proposed KNN-H.265 throughput varies from 1100 Mbps to 1110 Mbps when the PLR ranges from 0 to 12%. Hence, the proposed method gives higher throughput performance when compared with the existing methods.
Figure 6. Throughput vs. delay
Figure 6 represents the relation between throughput versus delay. Delay is an essential parameter in data transmission, especially for video signal transmission. The parameter delay depends on the channel parameters like propagation environment, reflection, distance, etc. When the delay is increased from 0 to 600 ms, the existing H.265 gives better throughput than the H.264 protocol. The existing two protocols are compared with the proposed KNN-H.265 protocol, the proposed protocol provides better throughput. At 300 m sec of delay, the proposed method improves 400 Mbps and 210 Mbps compared to the H.264 and H.265 protocols, respectively.
Figure 7. Distance vs. Delay
Figure 7 represents the relationship between distance versus delay. Delay is an essential parameter in data transmission and depends on the distance between the transmitter and the receiver. The parameter delay depends on the channel parameters like propagation environment, reflection, distance, etc. When the distance increases from 0 to 1000 meters, the existing H.265 gives less delay than the H.264 protocol. The existing two protocols are compared with the proposed KNN-H.265 protocol, the proposed protocol provides less delay/latency. At 500 meters of distance, the proposed method gives 410 m sec and 190 m sec less delay when compared with the H.264 and H.265 protocols, respectively.
- The article missed presenting the research novelty (Machine Learning). In the sense that they do not highlight what is missing from each of the other proposals. The authors should provide enough proof to convince the reader of superiority of the proposed schemes over the existing works.
Response: The authors applied the proposed LVT model with VIoT sensors to the modified H.265 protocol. The above figure shows the machine learning-based H.265 protocol containing KNN Classifier for the feature extractions by training and testing the samples. In general, H.265 protocol has video compression and extractors, but it contains low computational speed. Due to this, the latency will be increased, which causes the efficiency decrement. To improve the latency by reducing the computational rate, the authors proposed merging the KNN classifier in the H.265 protocol, shown in Figure 2.
Figure 2. KNN classifier-based H.265 protocol
The LVT and VIoT sensors are placed at the transmitter and receiver to process the video. The KNN-based H.265 protocol is applied to compress and extract faster in the network devices. The video transmission takes care of the LVT with compression and extraction, and the VIoT sensors for measuring the video quality. These two are computed based on the KNN-H.265 protocol. The main objective is to balance the process between the LVT and VIoT without any latency, as well as excellent video quality.
- The conclusion and future work part can be extended to have a better understanding of the approach and issues related to that which can be taken into consideration for future work.
Response:
The new Internet of Video Things refers to an emerging type of IoT system incorporating wireless visual sensors at the front end (VoIT) is introduced. H.265 coder with LVT programmable framework for simulating video communications in Wireless Visual Sensor Networks has experimented with error-tolerant methodologies such as block connecting. KNN classifier-based machine learning techniques with LVT and VoIT are applied to H.265 protocol for fast computation and transmission of the video data packets from transmitter to receiver. In this context, the proposed protocol performance is compared and analyzed with the existing H.264 and H.265 protocols. The parameters of throughput, delay, PSNR, Packet loss ratio, and SSIM are used to estimate the protocol's performance. The proposed protocol achieves improved performance by reducing latency up to 190 ms compared to the existing state of art H.264 and H.265 protocols; in terms of throughput proposed protocol achieves improved capacity compared with the H.264 and H.265 protocols at a nominal 300 ms of delay. Therefore, the performance of the proposed protocol can achieve better results when compared with the existing protocols in the case of lower latency and higher throughput. There is a scope to extend this work in the future by applying LSTM-based machine learning algorithms using KNN-H.265 protocol.

Round 2
Reviewer 2 Report
Although this version is improved significantly compared to the original version, it still has some errors that should be improved, as follows.
1) Title of the paper has some duplicate words, such as "Video Processing for Video Streaming Enhancement". You should revise it.
2) The Abstract should be rewritten more fluently,
such as "Simulation results for different QoE metrics concerning user-developed videos have been demonstrated, outperforming the existing metrics using MATLAB Software".
3) In line 106, no title of the Table.
4) Equations in lines 212 and 221 aren't numbered.
5) References [19-25] aren't cited.
Finally, in Section 1, the authors should mention the relationship between the research issue, IoT, and 5G/6G contexts. It will highlight your contributions, and cite some recent papers:
- "An efficient edge computing management mechanism for sustainable smart cities, Sustainable Computing: Informatics and Systems, Vol. 38, 100867, 2023.
- "Innovative Trends in the 6G Era: A Comprehensive Survey of Architecture, Applications, Technologies, and Challenges," IEEE Access, vol. 11, pp. 39824-39844, 2023.
Authors should perform a final strict and comprehensive revision, and double-check before submitting.
Author Response
- Title of the paper has some duplicate words, such as "Video Processing for Video Streaming Enhancement". You should revise it.
Response: As per reviewer’s suggestion title of the paper is revised and incorporated in the revised manuscript.
Optimized Visual Internet of Things for Video Streaming Enhancement in 5G Sensor Network Devices
2) The Abstract should be rewritten more fluently,
such as "Simulation results for different QoE metrics concerning user-developed videos have been demonstrated, outperforming the existing metrics using MATLAB Software".
Response: As per reviewer’s suggestion, Abstract is revised and incorporated in the revised manuscript.
The global expansion of the Visual Internet of Things (VIoT) deployment with multiple devices and sensor interconnection has been widespread. Frame collusion and buffering delays are the primary artifacts in the broad area of VIoT networking applications due to significant packet loss and network congestion. Numerous studies have been carried out on the impact of packet loss on Quality of Experience (QoE) for a wide range of applications. In this paper, a lossy video transmission framework for VIoT considering the KNN classifier merged with the H.265 protocols. The performance of the proposed framework was assessed considering the congestion on encrypted static images transmitted to the wireless sensor networks. The performance analysis of the proposed KNN-H.265 protocol is compared with the existing traditional H.265 and H.264 protocols. The analysis suggests that the traditional H.264 and H.265 protocols cause video conversation packet drops. The performance of the proposed protocol is estimated with the parameters of frame number, delay, throughput, packet loss ratio, and Peak Signal to Noise Ratio (PSNR) on MATLAB simulation software. The proposed model gives 4% and 6% better PSNR values than the existing two methods and better throughput.
- In line 106, no title of the Table.
Response: As per reviewer’s comment, title of the table is placed in the revised manuscript
- Equations in lines 212 and 221 aren't numbered.
Response: As per the reviewer’s suggestion, the equation numbers are assigned and cited in the text of revised manuscript.
- References [19-25] aren't cited.
Response: As per the reviewer’s suggestion, all references are cited in the revised manuscript.
- Finally, in Section 1, the authors should mention the relationship between the research issue, IoT, and 5G/6G contexts. It will highlight your contributions, and cite some recent papers:
- "An efficient edge computing management mechanism for sustainable smart cities, Sustainable Computing: Informatics and Systems, Vol. 38, 100867, 2023.
- "Innovative Trends in the 6G Era: A Comprehensive Survey of Architecture, Applications, Technologies, and Challenges," IEEE Access, vol. 11, pp. 39824-39844, 2023.
Response: As per the reviewer’s suggestion, the references are placed and cited in the revised manuscript and also the contributions are highlighted.
5G can offer high throughput with low latency by connecting many devices using IoT-based network infrastructure. Nowadays, 5 G-enabled IoT gives service to smart cities, healthcare, defence and education etc. One of the biggest tasks of 5 G-enabled IoT is to mitigate the service response time with computational techniques. The authors proposed the edge computing technique with IoT for smart city users. [32]
6G networking is considered to show more excellent performance than 5G with respect to reliability parameters, low delay and high bandwidth. 6G technology is developed by migrating the space, cellular, and underwater networks. [33]

Reviewer 3 Report
Although I appreciate seeing that the authors have tried to address my concerns, some of them still remain unresolved.
1. This paper should summarize those results and give a comprehensive performance comparison with previous works (Optimized). It needs to highlight the research main contribution with some brief indications and numerical improvement percentages in section of result.
2. An explanation of how the training and test data sets were organized from the overall dataset is required. Moreover, the results should include learning curves for training and test data sets.
3. More experiments and some comparisons with other up-to-date methods should be addressed or added to back your claims to expand your experiments and analysis of results further, such as [A].
[A] S. Islam, A. Budati, M. Hasan, S. Goyal and A. Khanna, Performance analysis of video data transmission for telemedicine applications with 5G enabled Internet of Things, Computers and Electrical Engineering, vol.108, 2023.
Author Response
Although I appreciate seeing that the authors have tried to address my concerns, some of them still remain unresolved.
- This paper should summarize those results and give a comprehensive performance comparison with previous works (Optimized). It needs to highlight the research main contribution with some brief indications and numerical improvement percentages in section of result.
Response: As per reviewer’s suggestion, the comprehensive performance comparison is placed in the revised manuscript. The numerical improvement percentages in results section are shown in the revised manuscript.
- An explanation of how the training and test data sets were organized from the overall dataset is required. Moreover, the results should include learning curves for training and test data sets.
Response: As per the reviewer’s comment, the data is generated using the own videos and the data is trained and tested with the KNN classifier-based machine learning model. We have generated the samples of 10000 and all the samples are trained and tested and also plotted the curves for various parameters.
- More experiments and some comparisons with other up-to-date methods should be addressed or added to back your claims to expand your experiments and analysis of results further, such as [A].
[A] S. Islam, A. Budati, M. Hasan, S. Goyal and A. Khanna, Performance analysis of video data transmission for telemedicine applications with 5G enabled Internet of Things, Computers and Electrical Engineering, vol.108, 2023.
Response: As per the reviewer’s suggestion, sensing error parameter is estimated and compared with the existing methods and also incorporated in the revised manuscript. As per the reference given in [A] the simulation results are done for the sensing error and placed in the revised manuscript.
Figure 8. Sensing Error
Figure 8 represents the relationship between distance versus probability of sensing error. The distance between the transmitter and receiver increases, and the sensing error increases. At 400 meters distance, the existing H.264 and H.265 gives 15% and 9% less sensing error than the proposed KNN-H.265 technique.

Round 3
Reviewer 2 Report
The version is improved significantly, In the reviewer's opinion, it can be considered for publishing.
The authors should minor revise typos/writing, as follows:
1) Line 106, "table 1. [26] " revise to "Table 1 [26].", The same in lines 111, 114, etc. You should double-check throughout the manuscript.
2) In Table 1, add the reference at the end of the cited author name, for example:
"Chen C.W. et al." revise to "Chen C.W. et al. [xx]", where, [xx] is the reference.
3) Line 217, "Eqn. (2) & (3)" revise to "Eq. (2) & (3)" or "Equations (2) & (3)".
4) 4) The references should be arranged based on the cited order.
Author Response
Response to Reviewer Comments
Comments and Suggestions for Authors
The version is improved significantly, In the reviewer's opinion, it can be considered for publishing.
The authors should minor revise typos/writing, as follows:
- Line 106, "table 1. [26] " revise to "Table 1 [26].", The same in lines 111, 114, etc. You should double-check throughout the manuscript.
Response: As per the reviewer’s suggestion, where ever the “table” is there, that is replace with “Table” in the revised manuscript.
2) In Table 1, add the reference at the end of the cited author name, for example:
"Chen C.W. et al." revise to "Chen C.W. et al. [xx]", where, [xx] is the reference.
Response: As per the reviewer’s suggestion, references are cited in the Table 1 of the revised manuscript
- Line 217, "Eqn. (2) & (3)" revise to "Eq. (2) & (3)" or "Equations (2) & (3)".
Response: As per the reviewer’s suggestion, the Eqn. is replaced with Eq. in the revised manuscript
- The references should be arranged based on the cited order.
Response: As per the reviewer’s suggestion, the references are rearranged in the cited order and also incorporated in the revised manuscript.

Reviewer 3 Report
Unfortunately, after two rounds of revisions, the authors still failed to appropriately address my concerns.
1. This paper should summarize the results and give a comprehensive performance comparison with previous works to prove the Optimized (mathematical).
2. The results should include learning curves for training and test data sets.
3. The authors did not provide solid achievements in this manuscript since this paper seems to be a somewhat incremental piece of work based on earlier research results [A].
[A] S. Islam, A. Budati, M. Hasan, S. Goyal and A. Khanna, Performance analysis of video data transmission for telemedicine applications with 5G enabled Internet of Things, Computers and Electrical Engineering, vol.108, 2023.
Author Response
Response to Reviewer Comments
- This paper should summarize the results and give a comprehensive performance comparison with previous works to prove the Optimized (mathematical).
Response: We have revised the manuscript by adding new comparative data analysis and graphical data interpretation. For example: We have provided the following results in Table 3, table 4, and table 6.
Table 3: comparative performance between frame number vs. PSNR
|
Frame number |
PSNR (dB) |
||
|
H.264 |
H.265 |
Proposed |
|
|
300 |
30 |
32 |
40 |
|
350 |
25 |
31 |
37 |
|
400 |
22 |
29 |
35 |
|
450 |
20 |
37 |
33 |
|
500 |
18 |
25 |
33 |
|
550 |
13 |
25 |
33 |
|
600 |
13 |
25 |
33 |
Table 4: comparative performance between PLR vs. Throughput
|
PLR |
Throughput (Mbps) |
||
|
H.264 |
H.265 |
Proposed |
|
|
2 |
492 |
802 |
1094 |
|
4 |
493 |
804 |
1096 |
|
6 |
497 |
806 |
1092 |
|
8 |
492 |
808 |
1096 |
|
10 |
490 |
808 |
1098 |
|
12 |
493 |
806 |
1096 |
Table 5: comparative performance between delay vs. Throughput
|
Frame number |
Throughput (Mbps) |
||
|
H.264 |
H.265 |
Proposed |
|
|
0 |
698 |
398 |
188 |
|
100 |
659 |
423 |
223 |
|
200 |
728 |
464 |
298 |
|
300 |
765 |
506 |
241 |
|
400 |
734 |
488 |
315 |
|
500 |
721 |
528 |
352 |
|
600 |
769 |
586 |
393 |
Table 6: comparative performance between distance vs. delay
|
Distance (meters) |
Delay (m sec.) |
||
|
H.264 |
H.265 |
Proposed |
|
|
0 |
698 |
388 |
180 |
|
200 |
649 |
413 |
213 |
|
400 |
718 |
476 |
288 |
|
600 |
755 |
545 |
331 |
|
800 |
834 |
601 |
415 |
|
1000 |
888 |
656 |
452 |
- The results should include learning curves for training and test data sets.
Response: We have revised the manuscript by including new results on learning curve for the training and test data set for the proposed method.
Figure 8. Learning curve for KNN model
We employed learning curve for analyzing the KNN model on H.265 protocol with LVT model detection. The learning curve of KNN model shows that the training and test converges together as we observe in Figure 8. Thus, we conclude that the performance of the KNN model tends to improves much with larger number of training samples. Moreover, the learning curve for KNN, demonstrated in figure 9, shows that the model is not suffering from overfitting or under fitting as both the training and test accuracy tends to increase with an increase in the number of observations.
- The authors did not provide solid achievements in this manuscript since this paper seems to be a somewhat incremental piece of work based on earlier research results [A].
[A] S. Islam, A. Budati, M. Hasan, S. Goyal and A. Khanna, Performance analysis of video data transmission for telemedicine applications with 5G enabled Internet of Things, Computers and Electrical Engineering, vol.108, 2023.
Response: As per the reviewer’s suggestion, the proposed research work is compared with the existing work [A] and incorporated in the revised manuscript.
The authors done the research in [A] is, they have proposed a single buffer instead of double buffer with Multi-Sensory (MS) alerts and estimated the performance of the MS-KNN classifier based single buffer with the double buffer system. The performance metrics are used to estimate are sensing error, collision error and propagation error etc. In this proposed research paper, the authors applied the LVT technique to KNN classifier enabled H.265 protocol for VIoT. The performance of the proposed method purely depends on the video signal transmission in the channel. So, the authors estimated the parameters like throughput, delay, packet loss ratio, frame number and PSNR etc. And also estimated the performance of the proposed LVT-KNN classifier with the mentioned parameters in [A] like sensing error, propagation error and collision error. The performance of the proposed method is compared with the existing MS-KNN single buffer [A] and identified the better performance method among it and shown in from figures 9 to 11.
Figure 9. Collision Error
Figure 9 represents the relationship between distance versus probability of collision error. The distance between the transmitter and receiver increases, and the collision error also varies. At 200 meters distance the number of users is more, huge number of video data packets transmission is required. Due to this huge volume of data transmission, collision errors are occurred. So, there is a more probability to get collision error is more than long distance. At this distance, MS-KNN single buffer (existing) gives 16% more collision error probability than proposed LVT-KNN classifier method.
Figure 10. Propagation Error
Figure 10 represents the relationship between distance versus probability of propagation error. The distance between the transmitter and receiver increases, and the propagation error also increases. Because the distance is increased, the signal strength is going to decrease and which causes to raise the propagation errors. At 400 meters distance, the MS-KNN single buffer (existing) gives 5% more propagation error probability than proposed LVT-KNN classifier method.
Figure 11. Sensing Error
Figure 11 represents the relationship between distance versus probability of Sensing error. The distance between the transmitter and receiver increases, and the sensing error also increases. At 400 meters distance, the MS-KNN single buffer (existing) gives 15% more sensing error probability than proposed LVT-KNN classifier method.
